# PHENSIM: Phenotype Simulator

**Salvatore Alaimo**[1]*, **Rosaria Valentina Rapicavoli**[1,2], **Gioacchino P. Marceca**[1], **Alessandro La Ferlita**[1,2], **Oksana B. Serebrennikova**[3], **Philip N. Tsichlis**[4], **Bud Mishra**[5], **Alfredo Pulvirenti**[1], **Alfredo Ferro**[1]*

1 Bioinformatics Unit, Department of Clinical and Experimental Medicine, University of Catania, Catania, Italy, 2 Department of Physics and Astronomy, University of Catania, Catania, Italy, 3 Molecular Oncology Research Institute, Tufts Medical Center, Boston, Massachusetts, United States of America, 4 Department of Cancer Biology and Genetics and the James Comprehensive Cancer Center, Ohio State University, Columbus, Ohio, United States of America, 5 Department of Computer Science, Courant Institute of Mathematical Sciences, New York University, New York, New York, United States of America

* salvatore.alaimo@unict.it (SA); alfredo.ferro@unict.it (AF)

**Data Availability Statement:** Source code for the PHENSIM algorithm is available at https://github.com/alaimos/mithril-standalone/tree/mithril-2.2. The source code for the web application is available at https://github.com/alaimos/phensim. All raw data, input files, and other source codes are

## Abstract

Despite the unprecedented growth in our understanding of cell biology, it still remains challenging to connect it to experimental data obtained with cells and tissues' physiopathological status under precise circumstances. This knowledge gap often results in difficulties in designing validation experiments, which are usually labor-intensive, expensive to perform, and hard to interpret. Here we propose PHENSIM, a computational tool using a systems biology approach to simulate how cell phenotypes are affected by the activation/inhibition of one or multiple biomolecules, and it does so by exploiting signaling pathways. Our tool's applications include predicting the outcome of drug administration, knockdown experiments, gene transduction, and exposure to exosomal cargo. Importantly, PHENSIM enables the user to make inferences on well-defined cell lines and includes pathway maps from three different model organisms. To assess our approach's reliability, we built a benchmark from transcriptomics data gathered from NCBI GEO and performed four case studies on known biological experiments. Our results show high prediction accuracy, thus highlighting the capabilities of this methodology. PHENSIM standalone Java application is available at https://github.com/alaimos/phensim, along with all data and source codes for benchmarking. A web-based user interface is accessible at https://phensim.tech/.

## Author summary

Despite the unprecedented growth in our understanding of cell biology, it still remains challenging to connect it to experimental data obtained with cells and tissues' physiopathological status under precise circumstances. This knowledge gap often results in difficulties in designing validation experiments, which are usually labor-intensive, expensive to perform, and hard to interpret. In this context, 'in silico' simulations can be extensively applied in massive scales, testing thousands of hypotheses under various conditions, which is usually experimentally infeasible. At present, many simulation models have

available for download at https://github.com/alaimos/phensim/tree/master/Benchmark.

**Funding:** ALF is supported by the Ph.D. fellowship on Complex Systems for Physical, Socio-economic and Life Sciences funded by the Italian MIUR "PON RI FSE-FESR 2014-2020". SA, AF, and AP have been partially supported by the following research project: PO-FESR Sicilia 2014-2020 "DiOncoGen: Innovative diagnostics." AF has also been partially supported by the research project "Experimental and Computational Study on Endogenous and Synthetic MicroRNA Folding with application in Oncology and Psychiatry (ECSMiRNAFOP)" funded by the University of Catania – "PIAno di inCEntivi per la RIcerca di Ateneo (PIACERI) 2020/2022". SA computational work has been partially supported by the Google Cloud Research Credits Program (Project Id: phensim). BM was supported by a National Cancer Institute Physical Sciences-Oncology Center Grant U54 CA193313-01 and a US Army grant W911NF1810427. The funders had no role in study design, data collection and analysis, decision to publish, or preparation of the manuscript.

**Competing interests:** The authors have declared that no competing interests exist.

become available. However, complex biological networks might pose challenges to their performance.

We propose PHENSIM, a computational tool using a systems biology approach to simulate how cell phenotypes are affected by the activation/inhibition of one or multiple biomolecules, and it does so by exploiting signaling pathways. We implemented our tool as a freely accessible web application, hoping to allow 'in silico' simulations to play a more central role in the modeling and understanding of biological phenomena.

This is a *PLOS Computational Biology* Methods paper.

## Introduction

Cells of living organisms are continuously exposed to signals originating in both the extracellular and the intracellular microenvironments. These signals regulate multiple cellular functions, including gene expression, chromatin remodeling, DNA replication and repair, protein synthesis, and metabolism. The proper response to signals depends on the expression, activation, or inhibition of sets of interrelated genes/proteins, acting in a well-defined order within the framework of vector-driven biological processes, aiming to reach specific endpoints. Such subcellular processes are referred to as biological pathways [1].

In this context, the study of genome and transcriptome, the definition of protein-protein interaction networks, and association studies between gene sets and molecular mechanisms in humans have produced valuable biological information. However, despite the improvements in our understanding of cell biology, it is challenging to link omics data to the physiopathological status of cells, tissues, or organs under specific conditions. Besides, studies addressing these issues are often labor-intensive, expensive to perform, and produce big datasets for analysis.

Recently, systems biology computational approaches have emerged as efficient means capable of bridging the gap between experimental biology at the system-level and quantitative sciences [2]. Indeed, such methods can be used as time- and cost-saving solutions for efficient *in silico* predictions [2,3]. Here, network analysis is playing a central role in modeling and understanding biological phenomena. In this perspective, simulation methodologies can help understand the intricate interaction patterns between molecular entities, significantly improving manual analysis. Furthermore, 'in silico' simulations can be extensively applied in massive scales, testing thousands of hypotheses under various conditions, which is usually experimentally infeasible.

At present, many simulation models have become available. However, they can be grouped into two broad categories: (i) discrete/logic or (ii) continuous models [4]. Discrete models represent each element's state in a biological network as discrete levels, and the temporal dynamic is also discretized. At each time step, the state is updated according to a function, determining how an entity's state depends on the state of other (usually connected) entities. Boolean networks [5,6] and Petri nets [7] represent two types of discrete models. BioNSi (Biological Network Simulator) [8] is an intuitive model, implemented as a Cytoscape 3 plugin [9]. It can use KEGG pathways [10] as a network model and represents each element in discrete states (usually up to 10). At each simulation time point, the state of a node is updated using an effect

function. The simulation ends as soon as it reaches a steady state. The model is easy to use. However, a more complex biological network might pose challenges to its performance.

Continuous models usually produce real continuous measurements instead of discretized values, simulating network dynamics over a continuous timescale. Although they could provide a greater degree of accuracy, these methods are limited by our current description of the biological systems and our measurement techniques' capabilities. Continuous linear models [11,12] and flux balance analysis [13] are the most representative continuous models.

Pathway modeling is an essential step for building networks that simulation methodologies can use. SBML is an open and interchange format for computer models of biological processes. However, converting pathways in annotated SBML files suitable for simulation models is not easy. Several tools such as KEGGconverter [14] or KENeV [15] have been specifically developed for this objective. These tools can also consider crosstalk with neighboring pathways, providing improved simulation accuracy. However, KEGGconverter has not been updated recently, and KENeV does not integrate post-transcriptional regulatory interactions or REACTOME pathways.

Here, we present PHENSIM (PHENotype SIMulator), a web-based, user-friendly tool allowing phenotype predictions on selected cell lines or tissues in 25 organisms, including models such as *Homo sapiens*, *Mus musculus*, *Rattus norvegicus*, and *Caenorhabditis elegans*. PHENSIM uses a probabilistic algorithm to compute the effect of dysregulated genes, proteins, microRNAs (miRNAs), and metabolites on KEGG and REACTOME pathways. Results are summarized through a Perturbation, which represents the expected magnitude of the alteration, and an Activity Score, which is an index of both the predicted effect of a gene dysregulation on a node (up- or down-regulation) and its likelihood. All values are also computed at the pathway-level. Moreover, to achieve greater accuracy, PHENSIM performs all calculations in the KEGG meta-pathway, obtained by merging all pathways [16] (see Methods) and integrates information on miRNA-target and transcription factor (TF)-miRNA extracted from online public knowledge bases [17]. Furthermore, the meta-pathway can be extended with REACTOME pathways to integrate a broader information source for cellular networks. We implemented our tool as a freely accessible web application at the following URL: https://phensim.tech/

## Results

To assess the performances of PHENSIM, we performed a comprehensive experimental analysis, as detailed in the "Experimental Procedure and Benchmarking" section. First, we built a benchmark composed of transcriptomics experiments performed on cell lines where a single gene was perturbed (knockdown, CRISPR, or transfection). Then, we quantitatively evaluated PHENSIM performance with an additional dataset containing experimental measurements of gene expression changes following drug treatment of a cell line [18]. Finally, in order to present some of the experiments that PHENSIM can perform, we ran four simulations as case studies and manually analyzed their results.

The benchmark was built by taking public GEO series of up-/down-regulation of single genes in cell lines. We acquired 22 GEO series further divided into 50 sets of samples (see Experimental Procedure and Benchmarking for more details). The sets were categorized based on the genes present in KEGG pathways (DS1 contains all sample sets where the up- or down-regulated gene was in KEGG; DS2 all the other samples). PHENSIM and BioNSi simulations were evaluated in terms of Accuracy, Positive Predictive Value (PPV), Sensitivity and Specificity for genes showing altered expression, and PPV and False Negative Rate (FNR) for the others.

Our results show that PHENSIM has an average accuracy of 0.6295 for the dataset in the first category and 0.3650 for the second category. Whereas BioNSi offers an average accuracy of 0.0640 and 0.0735 for the datasets in the first and second categories. Nevertheless, PHEN-SIM has higher PPV than BioNSi (0.6899 and 0.5075, respectively) in the first and second categories (PHENSIM = 0.7350, BioNSi = 0.3282). PHENSIM also shows a greater Sensitivity and Specificity to BioNSi. Furthermore, since PHENSIM can extend KEGG pathways with REAC-TOME, we performed the same tests on such an extended network, comparing the results before and after the integration. However, we could not evaluate BioNSi capabilities in this context since it could not load the extended network due to its size. In this setting, PHENSIM showed an average accuracy of 0.6437 with comparable PPV (0.6349) although lower Sensitivity (0.5416) and comparable Specificity (0.9854) for DS1. A slight decrease of performance can be observed for DS2 (Accuracy: 0.3291, PPV: 0.7571, Sensitivity: 0.7622, Specificity: 0.9716). S1 Table reports the detailed comparison in terms of average metrics.

To assess performance differences between the two systems for each dataset, we provide several graphs comparing each metric. In Fig 1, we summarize the DS1 datasets' results, and in Fig 2, we report the results from the DS2 datasets. In each graph, we detail a single metric: Positive Predictive Value (PPV), Sensitivity and Specificity for genes showing altered expression, and PPV and False Negative Rate (FNR) for the non-altered ones. On the x-axis, we have PHENSIM performance, while on the y-axis, we have BioNSi. Each dot represents a dataset. The black line marks the points where the two algorithms have the same performance. We summarize the comparisons before and after adding REACTOME pathways in S1 Fig for DS1 and S2 Fig for DS2. In these graphs, the x-axis represents the PHENSIM performance with REACTOME, while on the y-axis, we have PHENSIM without REACTOME.

Moreover, to quantitatively evaluate network perturbation prediction, we chose an additional dataset of protein expression measurements following drug treatment of a cell line [18]. The dataset contains measurements of 124 protein levels in a time series from 10 minutes to 67 hours (8 timepoints). The authors followed the perturbation caused by the administration of 54 drug combinations, including several gene inhibitors (MEKi, AKTi, STAT3i, SRCi, mTORi, BETi, PKCi, RAFi, and JNKi). In Fig 3, we report the analysis results comparing PHENSIM steady-state predictions with each time point in terms of the Pearson Correlation Coefficient. Results show that PHENSIM predictions are coherent with the proteomics experiments, reaching the maximal correlation at 24h and 48h.

Finally, to complete our assessment of PHENSIM capabilities, we run several simulations to perform 4 case studies on known biological experiments: (i) anti-cancer effects of metformin, (ii) Everolimus (RAD001) treatment in breast cancer, (iii) effects of exosomal vesicles on hematopoietic stem/progenitor cells (HSPCs) in the bone marrow (BM) and (iv) testing TNFα/siTPL2-dependent synthetic lethality on a subset of human cancer cell lines. We examined the ability of PHENSIM to correctly predict the activity status of both individual genes/proteins and signaling pathways by comparing PHENSIM predictions with experimental data. In the following sections, we briefly report the results of two case studies: the anti-cancer effects of metformin and the testing TNFα/siTPL2-dependent synthetic lethality on a subset of human cancer cell lines. Detailed descriptions of all case studies are provided in S1 Text.

## Anti-cancer effects of metformin

Metformin is a widely prescribed agent for the treatment of type 2 diabetes [19–22]. It inhibits glucose production in the liver and increases insulin sensitivity in the peripheral tissues. Furthermore, metformin treatment reduces insulin secretion by β-pancreatic cells. The key molecule that executes these functions is AMP-activated protein kinase (AMPK). Several evidence

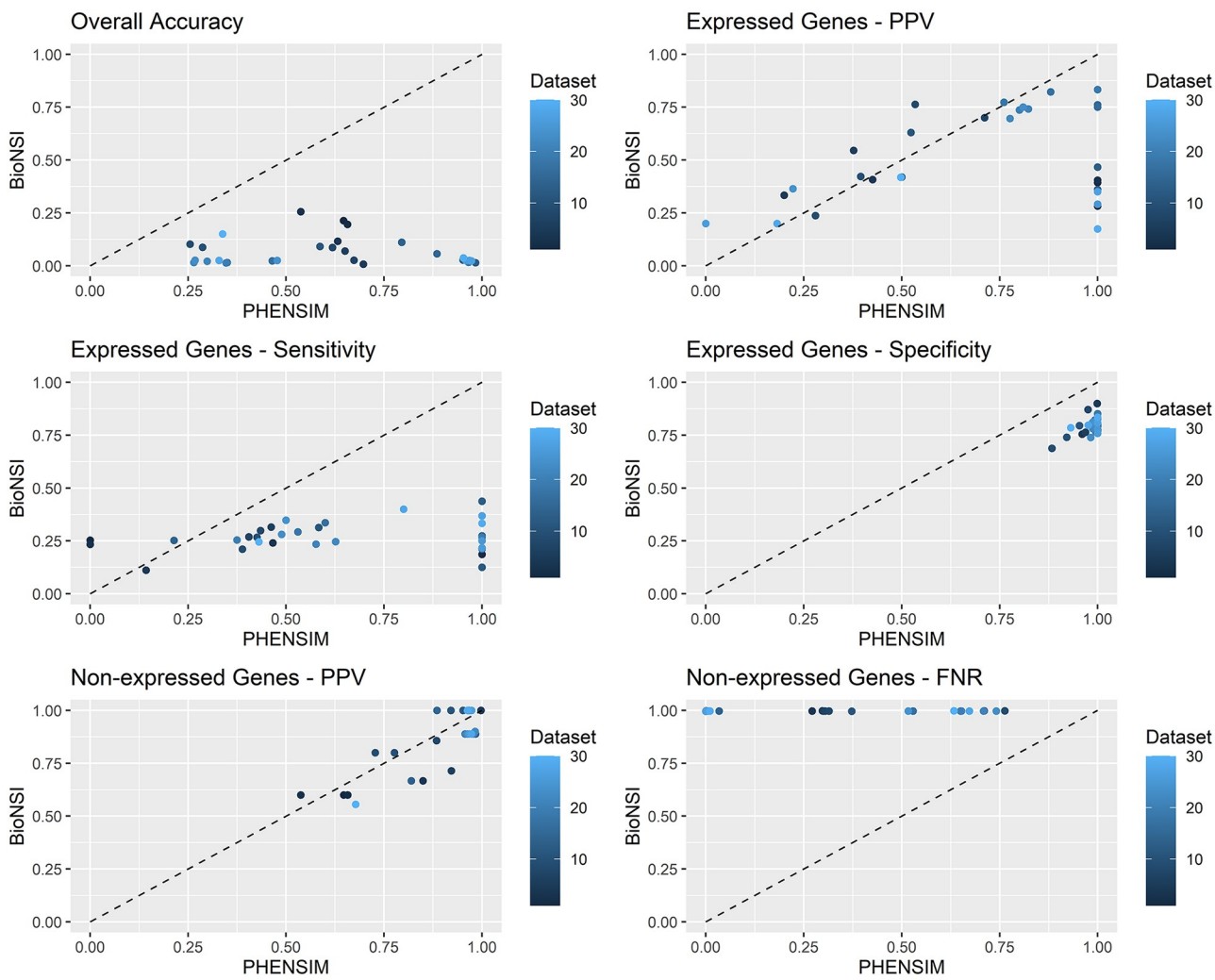

**Fig 1. Comparison between PHENSIM and BioNSi for datasets where the altered gene was in the meta-pathway.** Each graph reports one metric: Positive Predictive Value (PPV), Sensitivity and Specificity for genes showing altered expression, and PPV and False Negative Rate (FNR) for the non-altered ones. On the x-axis, we report PHENSIM performance, while on the y-axis, we present BioNSi. Each dot represents a dataset. The black line marks the points where the two algorithms have the same performance. On a dataset below the line, PHENSIM has better performance than BioNSi; above the line, it is the opposite.

indicates that metformin may also possess anti-cancer effects, especially in diabetic patients [19–21]. One of its major drivers seems to be the LBK1-AMPK signaling pathway [21]. An overview of the metformin-mediated effects is reported in Fig 4.

We ran PHENSIM to simulate the simultaneous upregulation of LKB1 and the downregulation of both insulin (Ins), IGF1, and GPD1 [23]. As expected, PHENSIM returned significant downregulation of Insulin and mTOR signaling (Insulin activity score = -8.7121, p-value 0.105; mTOR activity score = -8.7121, p-value 0.107). PI3K (phosphoinositide 3-kinase), AKT (serine/threonine-protein kinase Akt), and metabolite PIP3 (phosphatidylinositol (3,4,5)-triphosphate) were also downregulated. We also predicted the negative regulation of mTOR (perturbation = -0.00002) and the activation of the *repressor of translation initiation* 4EBP (perturbation = 0.0009). PHENSIM also predicted the inhibition of downstream nodes involved in protein synthesis as S6Ks (S3A Fig).

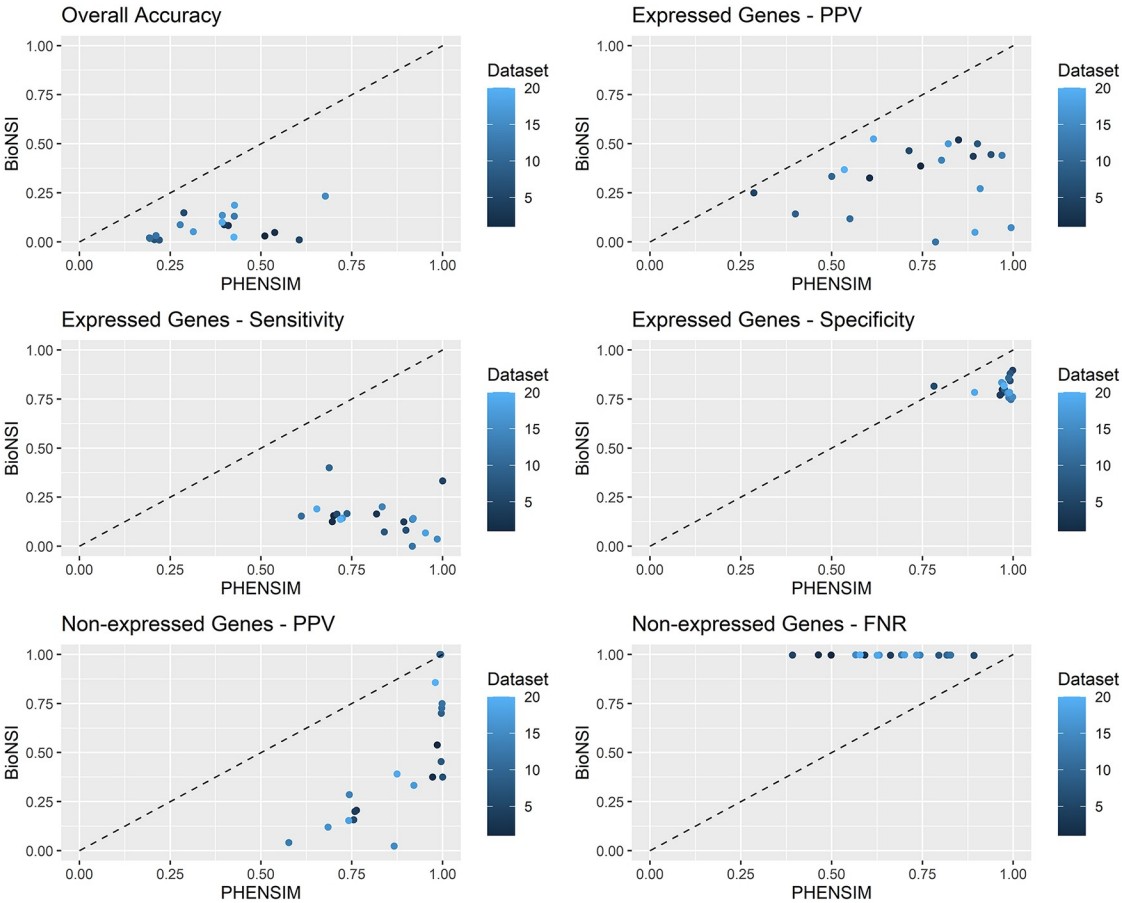

**Fig 2. Comparison between PHENSIM and BioNSi for datasets where the altered gene was not in the meta-pathway.** Each graph reports one metric: Positive Predictive Value (PPV), Sensitivity and Specificity for genes showing altered expression, and PPV and False Negative Rate (FNR) for the non-altered ones. On the x-axis, we report the PHENSIM performance, while on the y-axis, we have BioNSi. Each dot represents a dataset. The black line marks the points where the two algorithms have the same performance. On a dataset below the line, PHENSIM has better performance than BioNSi; above the line, it is the opposite.

MAPK (activity score = -8.7121, p-value 0.113, perturbation = -3.193292579) and NF-$\kappa$B (NF-$\kappa$B perturbation = -0.0008) signaling were predicted downregulated. Furthermore, several downregulated enzymes and metabolites were correctly detected by PHENSIM (S3B Fig).

## Testing TNFα/siTPL2-dependent synthetic lethality on a subset of human cancer cell lines

TNFα (tumor necrosis factor alpha), a type II transmembrane protein, is a member of the tumor necrosis factor cytokine superfamily and has an essential role in innate immunity and inflammation.

Although it can induce cell death, most cells are protected by a variety of rescue mechanisms.

In a recent paper, Serebrennikova et al. [24] showed that TPL2 (MAP3K8) is one of the TNFα-induced cell death checkpoints. Its knockdown resulted in the downregulation of miR-21 and the upregulation of its target CASP8 (caspase-8). This response, combined with the downregulation of caspase-8 inhibitor cFLIP (FADD-like IL-1β-converting enzyme inhibitory protein), resulted in the activation of caspase-8 by TNFα and the initiation of apoptosis

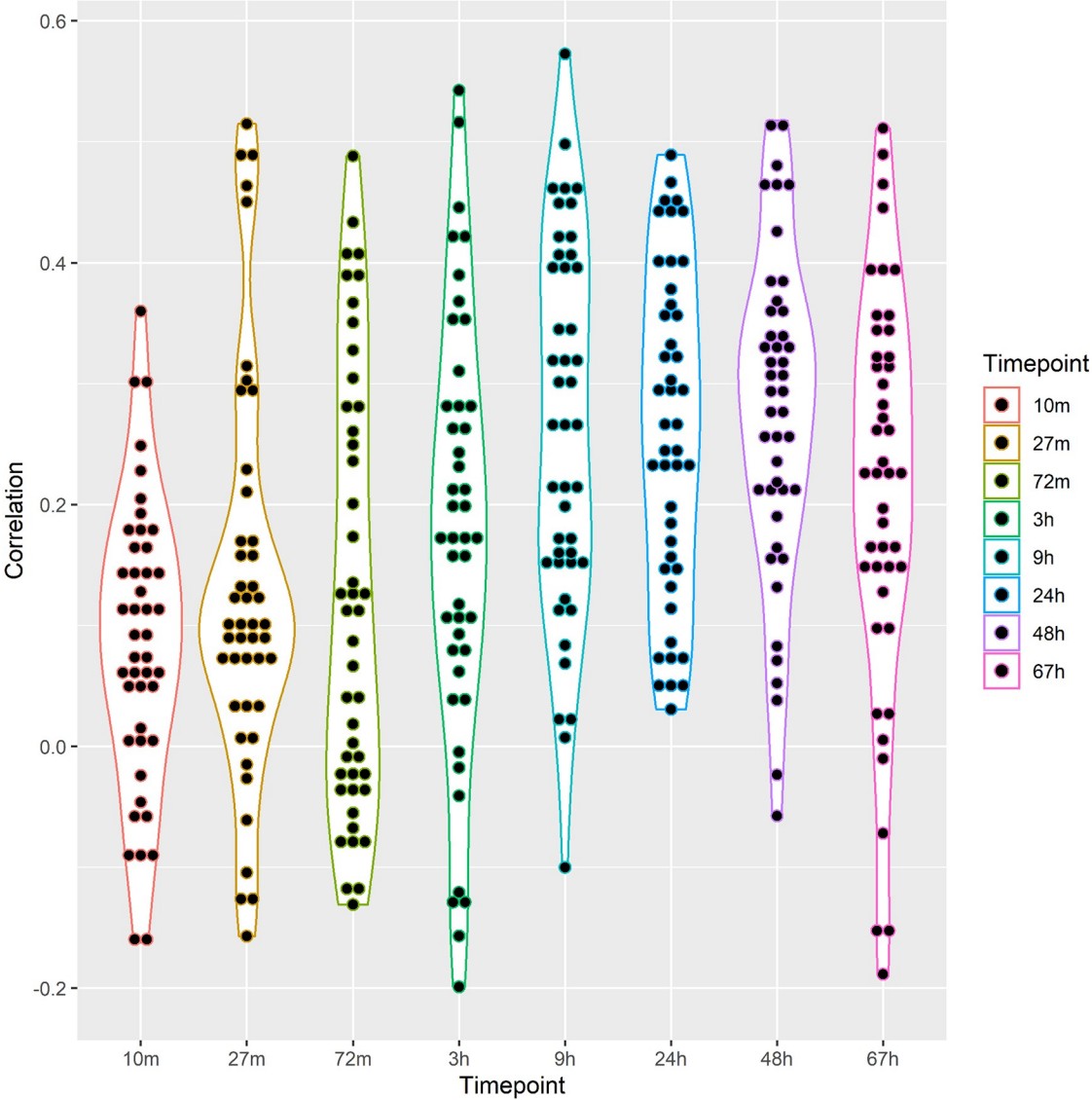

**Fig 3. Comparison between PHENSIM predictions and the proteomics measurements of Nyman et al.** [18]**.** We report the Pearson Correlation Coefficient computed between PHENSIM and the proteomics measurements for each timepoint and drug combination. Results are summarized through a violin plot detailing both the distribution and the values' density.

(Fig 5). The activation of caspase-8 also promotes the activation of the mitochondrial pathway of apoptosis. It is worth noticing that the activation of the apoptotic (caspase-8-dependent) pathway in TNFα/siTPL2 treated cells was observed in some but not all cancer cell lines, suggesting that correct prediction will depend on whether the data analyzed by PHENSIM are derived from sensitive or resistant cells.

To start the simulation, we set TPL2 and miRNA-21-5p as downregulated and TNFα as upregulated. Since our goal was to simulate the outcome of such treatment in six different cell lines (HeLa, HCT116, U2-OS, CaCo-2, RKO, and SW480), we ran six simulations. Each simulation had a diverse list of non-expressed genes, one for each cell line.

Among these tumor cell lines, only HeLa, HCT116, U2-OS were sensitive to treatment with TNFα/siTPL2. PHENSIM couldn't predict the upregulation of caspase-8 and the

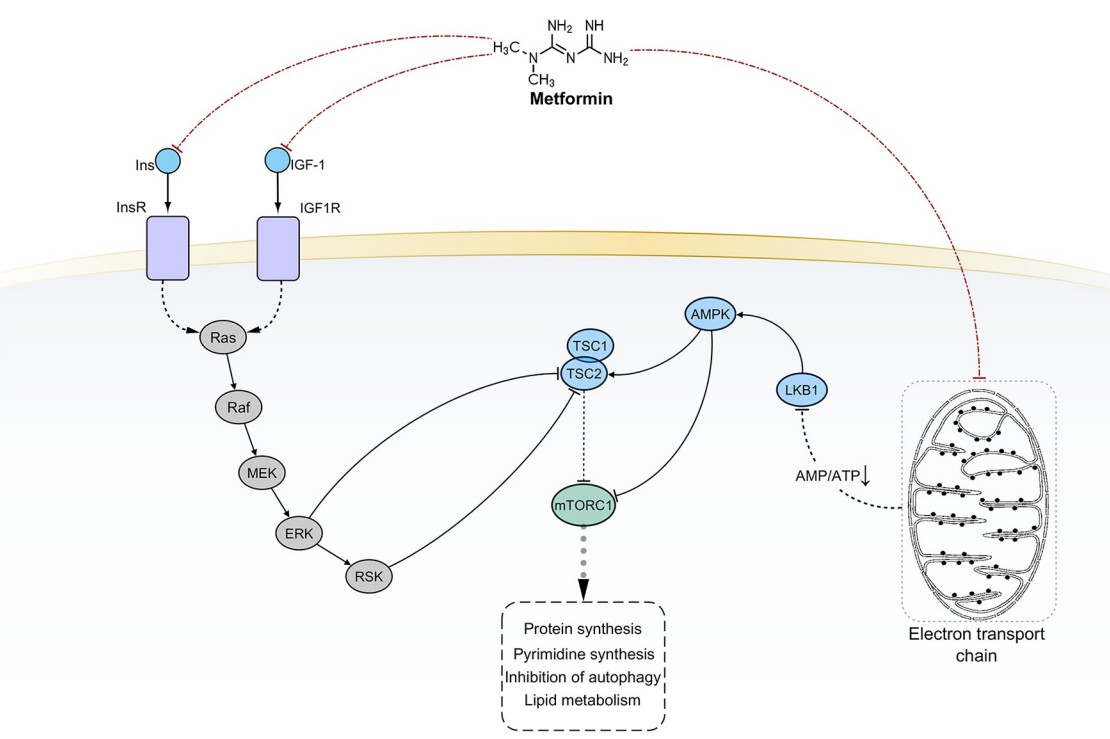

**Fig 4. The current model of metformin-mediated pharmacological effects.** Black solid edges represent direct interaction between first neighbor nodes. Dashed edges represent indirect interactions between nodes. Red dot-dashed edges evidence scientifically validated interactions considered for PHENSIM prediction.

downregulation of cFLIP for the six cell lines. PHENSIM did not predict any activity score for MCL1 (Mcl-1 apoptosis regulator) and XIAP (X-linked inhibitor of apoptosis).

PHENSIM could not predict the upregulation of the apoptosis inhibitors BCL2 and BCL-XL in all cell lines except for HCT116, where BCL2 results positively perturbated (perturbation = 0.001). PHENSIM also showed a negative perturbation of the inducer of mitochondrial apoptosis BAX only in HCT116 among the sensitives cell lines (S4 Fig).

Although these results do not entirely reflect our expectations as there are discrepancies between the in vitro experiment and our predictions, it was confirmed by results obtained in Serebrennikova et al. [24] that the change in the expression of such molecules was due to the activation of feedback mechanisms. Interestingly, this result was obtained only for four out of six cancer cell lines, of which three were sensitive (HeLa, HCT116, and U2-OS), and one was resistant (CaCo-2).

Furthermore, phosphorylated ERK, MEK, JNK, and p38 activity were strongly downregulated for all cell lines except for RKO, where PHENSIM predict only ERK and p38, and for Caco-2 cells, which result in a negative activity score for ERK and a weak perturbation for JNK and p38 genes. Finally, PHENSIM could not predict cIAP2 (baculoviral IAP repeat containing 2) activity, although we could observe a weak negative perturbation in RKO, as confirmed by the experimental data (S4A and S4B Fig).

## Discussion

This paper introduces PHENSIM, a flexible, user-friendly pathway-based simulation technique, and an in silico tool based on it. PHENSIM has been mainly developed to predict the effects of one or multiple molecular deregulations on cell/tissue phenotype. Thus, we view

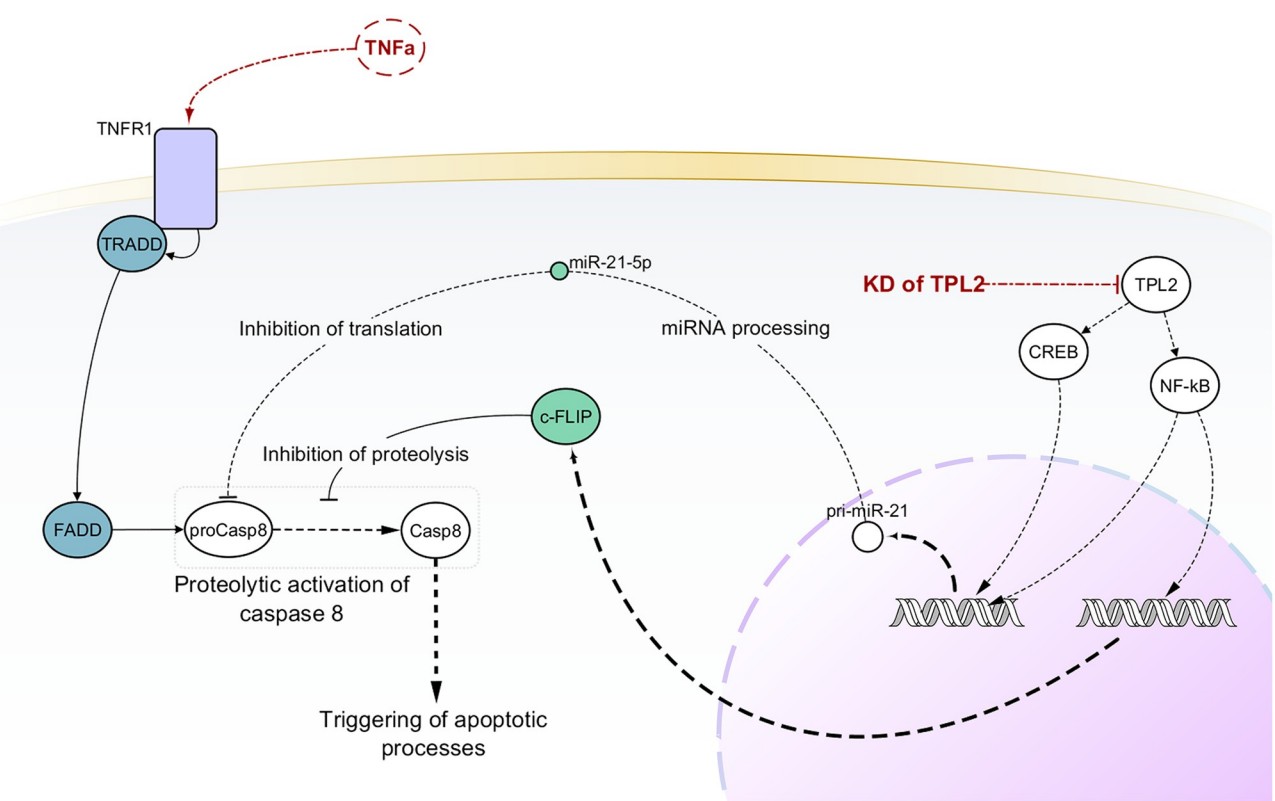

**Fig 5. Generalized model showing molecular mechanisms underlying the TNFα/siTPL2-dependent synthetic lethality.** Black solid edges represent direct interaction between first neighbor nodes. Dashed edges represent indirect interactions between nodes. Red dot-dashed edges evidence scientifically validated interactions considered for PHENSIM prediction.

PHENSIM as an easy-to-use, supportive pathway-based method that can make predictions of in vitro experiments targeting the expression of signaling processes' activity.

To evaluate our tool's potential, we built a benchmark of 50 case/control sample sets derived from 22 GEO series. Each set contained expression data of experiments regarding the up- or down-regulation of one single gene in a specific cell line. As previously described, 30 sample sets were directly used since the tested gene was already in KEGG. The remaining 20 sets were simulated through their differentially expressed genes (DEGs). Here, the main idea is that the DEGs can summarize the downstream alterations caused by the experiment. We compared our approach's performance with BioNSi, a Cytoscape plugin for modeling biological networks and simulating their dynamics. Results-based comparative evaluations were performed in terms of accuracy, Positive Predictive Value (PPV), Sensitivity and Specificity for genes showing altered expression, and PPV and False Negative Rate (FNR) for the non-altered ones.

We show that, on average, our tool obtains better results than BioNSi in terms of accuracy, PPV, Sensitivity, Specificity, and FNR. More in detail, for the 30 samples of DS1, we show that only in 11 cases BioNSi achieves a greater PPV than PHENSIM. However, Sensitivity and Specificity are still higher for our methodology. In the other 20 samples, PHENSIM consistently outperforms BioNSi. Furthermore, when looking at non-expressed genes, BioNSi has significantly higher FNR than PHENSIM.

Since PHENSIM can be easily extended with other pathway data sources, we integrated REACTOME pathways in our knowledge base and performed the same experiments.

However, we could not perform any comparison with BioNSi since it could not load the extended network. Results show that although we have a decreased accuracy, the overall Sensitivity and Specificity of the method are comparable or higher. Therefore, we can hypothesize that integrating more provenance sources for cellular networks will positively impact the results generated by PHENSIM.

Moreover, we quantitatively evaluated network perturbation prediction using a dataset of protein expression measurements following drug treatment [18]. Results show that PHENSIM predictions are coherent with the proteomics experiments, reaching the maximal correlation at 24h and 48h.

To further explore PHENSIM capabilities, we performed four case studies in different scenarios: drug administration to cultured cells (simulations 1 and 2), effects of exosomal-derived miRNAs in recipient cells (simulation 3), and the combined targeting of two signaling molecules, which are known to induce synthetic lethality in a subset of cell lines (simulation 4). After comparison, the literature data and PHENSIM predictions were in almost full agreement with simulation #1 and partial agreement with the three remaining simulations, showing a discrete degree of accuracy.

Discrepancies with baseline data suggest some limitations in the predictive potential of our method. However, since pathway analysis relies on prior knowledge about how genes, proteins, and metabolites interact, we hypothesize that such a negative outcome is at least partly due to the incompleteness of the existing knowledge employed in the study. Indeed, since the biological pathways on current databases are still largely fragmented, calculations based on them will inevitably produce less than ideal results [25]. One example of this limitation is provided by mTORC1 downstream signaling. It is known that mTORC1 promotes protein synthesis by phosphorylating p70SK and 4EBP. It also stimulates ribosome biogenesis via inhibitory phosphorylation of the RNA Polymerase III repressor MAF1 [26]. mTORC1-induced pyrimidine biosynthesis is stimulated by p70S6K-mediated phosphorylation of the CAD enzyme (carbamoyl-phosphate synthetase 2, aspartate transcarbamylase, and dihydroorotase). Furthermore, the upregulation of 5-phosphoribosyl-1 pyrophosphate (PRPP) is an allosteric CAD activator [27,28].

KEGG Pathways do not consider such interactions. Therefore, our tool could not predict any perturbations for these biological processes. Similar observations can be made for the downregulation of cFLIP in the siTPL2/TNFα-resistant cell lines by our method. However, we were able to identify indirect evidence of such activity. On the other hand, the correct predictions obtained for autophagy, RNA transport, and mTOR signaling in simulation 2, and the mitochondrial apoptotic pathway activation in simulation 4, suggest that, provided with the right information, PHENSIM is likely to obtain significantly better results.

A further limitation for pathway analysis methods is the current knowledge-base inability to contextualize gene expression and pathway activation in a cell- and condition-specific manner [25]. Furthermore, pathways do not consider protein isoforms encoded by different genes or differently processed mRNAs derived from a single gene. This poses a significant limitation since such isoforms may have unique and sometimes opposite signaling properties. By developing a strategy that allows removing non-expressed genes from the computation, we offer the user the possibility to contextualize predictions in a cell- or tissue-dependent manner. In conjunction with this, integrating KEGG pathways with information from post-transcriptional regulators such as miRNAs increased the results' accuracy, leading to considerable improvements in predictions [29]. Moreover, using the meta-pathway approach, instead of single disjointed pathways, partially addresses pathway independence [25].

In conclusion, PHENSIM showed good accuracy in most applications and could predict the effects of several biological events starting from the analysis of their impact on KEGG. We

believe that several discrepancies can be traced to the incompleteness of knowledge in KEGG pathways or the lack of appropriate cell- and condition-specific information. Such incompleteness can be partially addressed through a manual annotation of the pathways with the missing elements and links, including miRNA-target and TF-miRNA interactions. Furthermore, we plan to add other pathway databases such as Reactome or NCI pathway to enhance our meta-pathway. PHENSIM is limited to the simulation of changes in the expression or activity of signaling molecules. It is not suitable to simulate genetic aberrations unless they affect molecules' expression or activity directly. Despite these limitations, our approach shows appreciable utility in the experimental field as a tool for the reliable prioritization of experiments with greater success chances.

## Methods

### Overview of the method

PHENSIM is a randomized algorithm to predict the effect of (up/down) deregulated genes, metabolites, or microRNAs on the KEGG meta-pathway [16]. The meta-pathway is a network obtained by merging all KEGG pathways through their common nodes. This approach allows us to consider pathway crosstalk and, ideally, gives a more comprehensive representation of the human cell environment. Furthermore, the KEGG meta-pathway is annotated with experimentally validated miRNA-target and Transcription Factor-miRNA interactions to consider post-transcriptional expression modulation.

Currently, our method uses all KEGG pathways (downloaded on April 2020) with details on validated miRNA-targets inhibitory interactions downloaded from miRTarBase (release 8.0) [30] and miRecords (updated to April 2013) [31], and TF-miRNAs interactions obtained from TransmiR (release 2.0) [32]. Furthermore, since the method's architecture is easily extensible, we include the possibility of integrating REACTOME pathways to the meta-pathway environment, yielding a richer and more comprehensive model.

To start a simulation, PHENSIM requires a set of nodes (at least one) together with their "deregulation type" (up-/down-regulation) as input values. We can also provide: (i) a list of non-expressed genes, (ii) a set of new nodes or edges that will be added to the meta-pathway, and (iii) the organism. For the sake of clarity, we first define the case when input elements are independently altered. That is, input nodes whose expression is independently changed from one another (i.e., transfection of two siRNAs for knockdown of two genes). Next, we report an efficient and reliable technique to deal with dependent alterations.

PHENSIM uses the input to compute synthetic Log-Fold-Changes (LogFC) values. These values are then propagated within biological pathways using the MITHrIL algorithm proposed in Alaimo et al. 2016 [17] to establish how these local perturbations can affect the cellular environment. This propagation result is called a "Perturbation," reflecting the change of expression for a gene in a pathway (negative/positive for down-/up-regulation). This value is computed for each gene in the meta-pathway. Finally, PHENSIM summarizes all results using two values for each gene: the "*Average Perturbation*" and the "*Activity Score*" (*AS*). The average perturbation is the mean for all perturbation values computed during the simulation process and reproduces the expected change of expression for the entire process. The function of the Activity Score is twofold. The sign gives the type of predicted effect: positive for activation, negative for inhibition. The value is the log-likelihood that this effect will occur. Together with the AS, PHENSIM also computes a p-value through a bootstrapping procedure. All p-values are then corrected for multiple hypotheses using the q-value approach [33]. PHENSIM p-values are used to establish how biologically relevant the predicted alteration is for the simulated phenomena—i.e., the lower is a node p-value, the less likely it is that such alteration will occur by

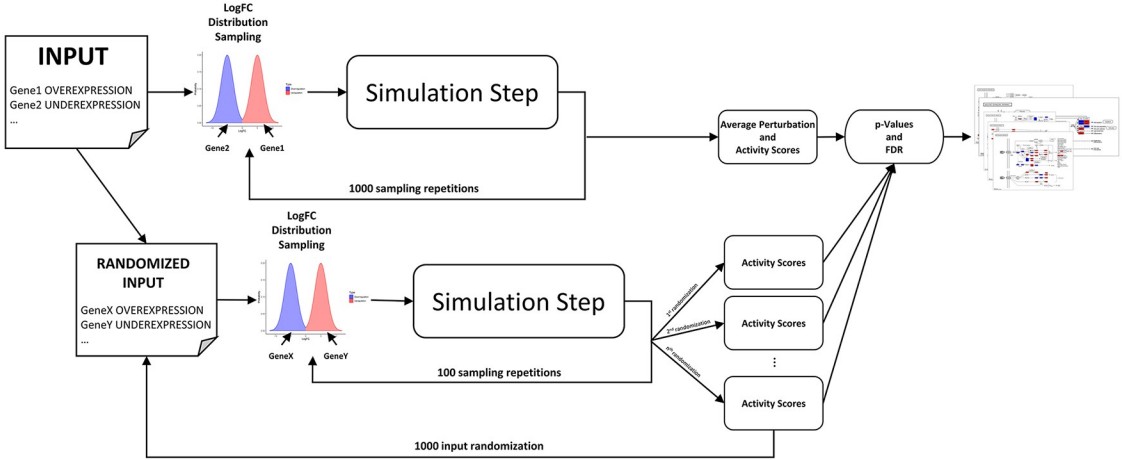

**Fig 6. Description of the PHENSIM algorithm.** First, the user provides a set of genes and the type of alteration (over-/under-expression). Then, synthetic LogFCs are generated, and a simulation step is performed. This procedure is repeated 1000 times to compute the *Activity Scores*. Next, user input is randomized, and 100 synthetic LogFC are generated to estimate *Activity Scores* using the simulation step. This input randomization is repeated 1000 times for greater precision. Finally, p-values are computed, and the False Discovery Rate is estimated using the q-value methodology.

chance. An overview of the PHENSIM algorithm is depicted in Fig 6. The algorithm comprises of 5 main steps. Given a user input, (i) synthetic LogFC are generated and a (ii) simulation step is performed. These steps are repeated 1000 times to (iii) compute the *AS*. Next, user input is (iv) randomized, and 100 synthetic LogFC are generated to estimate *AS* using the simulation step. The input is randomized 1000 times to obtain greater precision. Finally, (v) p-values are computed, and the False Discovery Rate is estimated using the q-value methodology.

PHENSIM is implemented as a Java application for easy deployment on multiple operating systems. The source code is included in the MITHrIL platform and available at https://github.com/alaimos/mithril-standalone/tree/mithril-2.2. A web application is also available at https://phensim.tech/. All experimental data and source codes generated or analyzed during this study are available at https://github.com/alaimos/phensim.

## Synthetic LogFC generation

PHENSIM relies on MITHrIL perturbation analysis to compute the state of a node in the KEGG meta-pathway. Starting from LogFCs, MITHrIL propagates them through the network to estimate node and pathway perturbation. Hence a critical step in the PHENSIM simulator is the generation of Synthetic LogFCs.

By analyzing experimental data from "The Cancer Genome Atlas (TCGA)," we infer the space of feasible LogFCs. First, we got all cancer and control samples of TCGA to compute LogFCs of each gene for each cancer sample. With these data, we then fit two normal distributions for positive and negative LogFCs, respectively. This analysis produced two normal distributions with a mean of 5 for up-regulation (-5 for down-regulation) and a standard deviation of 2.

Synthetic LogFCs are estimated by sampling the two distributions. More precisely, let $x \in \{-1,0,1\}$ be an input value, where $-1$ represents downregulation, $+1$ upregulation, and 0 no expression. At each simulation step, we generate a standard gaussian pseudorandom number,

$r_{\mathcal{N}}(x)$, by using the polar method [34]. Synthetic LogFCs are then computed as:

$$
LFC(x) = \begin{cases} \max\big(0, 2 * r_{\mathcal{N}}(x) + 5\big) & if \ x = 1 \\ \min\big(0, 2 * r_{\mathcal{N}}(x) - 5\big) & if \ x = -1 \\ 0 & if \ x = 0 \end{cases} \tag{1}
$$

## PHENSIM simulation step

Let the meta-pathway be defined as a graph $G(V, E)$ where $V = \{V_1, V_2, \ldots, V_m\}$ is the set of all biological elements (genes, metabolites, miRNAs), and $E \subset V \times V$ is the set of activating or inhibiting interactions. Moreover, without loss of generality, we define PHENSIM input $\mathcal{I} = \{V_1 = v_1, \ldots, V_n = v_n\}$ where $v_k \in \{1, 0, -1\}$ for $1 \leq k \leq n$, and $n \leq m$. As previously described, we represent downregulation with −1, upregulation with +1, and no expression with 0.

To compute the activity of a biological element, each node $V_i$ is considered as a discrete random variable that can assume three possible values: activated (1), inhibited (-1), or unchanged (0).

Given the input, the probability distribution of each variable is unknown. Therefore, we try to estimate it by generating synthetic LogFCs, which are then employed by MITHrIL perturbation analysis. Indeed, MITHrIL perturbation reflects the expected gene expression change when an alteration (expressed in terms of LogFC) is applied to a set of elements in the meta-pathway. Therefore, we collect these details to estimate a probability distribution empirically.

More in detail, given an input $\mathcal{I}$, at each step $t$ of the simulation, we compute a set of LogFCs, $\Delta E_{\mathcal{I}}(k, t) \ for \ 1 \leq k \leq m$, where:

$$
\Delta E_{\mathcal{I}}(k, t) = \begin{cases} 0 & if \ V_k \notin \mathcal{I} \\ LFC(v_k) & if \ V_k \in \mathcal{I} \end{cases}. \tag{2}
$$

Next, for each node $0 \leq i \leq m$, we estimate perturbation at step $t$ as:

$$
\mathcal{P}_{\mathcal{I}}(i, t) = \Delta E_{\mathcal{I}}(i, t) + \sum_{u \in U(i)} \frac{w(u, i)}{\sum_{d \in D(i)} w(u, d)} \mathcal{P}_{\mathcal{I}}(u, t), \tag{3}
$$

where $U(k)$ and $D(k)$ are the set of upstream and downstream nodes of $V_k$, respectively, and $w(j,k)$ is a weight reflecting the type of interaction between nodes $V_j$ and $V_k$. In PHENSIM, we use $w(j,k) = 1$ for all activating interactions, $w(j,k) = -1$ for all inhibiting ones. Finally, perturbations are returned for the computation of the *Activity Scores*. A detailed graphical representation of the calculation of Eq 3 is depicted in S5 Fig.

## Activity score computation

Given the input $\mathcal{I}$, *PHENSIM* summarizes the activity of a node $V_i$ in an A*ctivity Score*, $\mathcal{A}_{\mathcal{I}}(i)$. The function of the *AS* is twofold. The sign gives the type of predicted effect: positive for activation, negative for inhibition. The value is the log-likelihood that such a result will occur. Therefore, to determine its value, we need to estimate the probability distribution of each node. To this end, we repeat the simulation step $\mathcal{T}$ times to compute a set of perturbations $\mathcal{P}_{\mathcal{I}}(i) = \{\mathcal{P}_{\mathcal{I}}(i, t) \ where \ 1 \leq t \leq \mathcal{T}\}$ for each node $V_i$ of the graph.

Since the perturbation is negative for downregulation, positive for upregulation, and 0 for no alteration, we can use the sign function to determine node state. Therefore, by counting the number of times each state appears during the simulation, we can empirically estimate the

probability $Pr\ (V_i = v_i | \mathcal{I})$ for $1 \leq k \leq m$ as:

$$Pr(V_i = v_i | \mathcal{I}) = \frac{\{p \in \mathcal{P}_{\mathcal{I}}(i) | sign(p) = v_i\}}{\mathcal{T}}\ . \tag{4}$$

Finally, the activity score for a node $V_i$ can be determined as:

$$\mathcal{A}_{\mathcal{I}}(i)\ =\ \begin{cases} \log_2\left(\dfrac{Pr(V_i = 1 | \mathcal{I})}{1 - Pr(V_i = 1 | \mathcal{I})}\right) & \text{if } Pr(V_i = 1 | \mathcal{I}) > (1 - Pr(V_i = 1 | \mathcal{I})) \\[2mm] -\log_2\left(\dfrac{Pr(V_i = -1 | \mathcal{I})}{1 - Pr(V_i = -1 | \mathcal{I})}\right) & \text{if } Pr(V_i = -1 | \mathcal{I}) > (1 - Pr(V_i = -1 | \mathcal{I})) \\[2mm] 0 & \text{if } Pr(V_i = 0 | \mathcal{I}) > (1 - Pr(V_i = 0 | \mathcal{I})) \end{cases} . \tag{5}$$

In all our experiments, we set $\mathcal{T} = 1000$ for the simulation step. A detailed graphical representation of the computation of Eqs 4 and 5 is depicted in S5F Fig.

## Bootstrapping and randomization

With the Activity Score, PHENSIM computes a p-value to establish which of the observed alterations are biologically relevant and not obtained by chance. Our idea is that a node is biologically relevant for the input if it is unlikely to observe a similar alteration when perturbing random nodes in the same way. We achieve this through a bootstrapping procedure together with input randomization. Given the $\mathcal{I} = \{V_1 = v_1, \ldots, V_n = v_n\}$, we compute $\mathcal{R}$ random input set by taking arbitrary nodes from the KEGG meta-pathway. That is, for each randomization $1 \leq r \leq \mathcal{R}$, we define a random input set $\mathcal{I}_{\mathcal{R}}(r) = \left\{V_{j_1(r)} = v_1, \ldots, V_{j_n(r)} = v_n\right\}$ where $V_{j_k(r)} \in V$ is a node of the meta-pathway chosen randomly in $V$. Next, for each input set, we compute synthetic LogFCs and run $\mathcal{T}$ simulation steps to determine random *Activity Scores*, $\mathcal{A}_{\mathcal{I}_{\mathcal{R}}(r)}(i)$. For the bootstrapping and randomization procedures, we set $\mathcal{R} = 1000$ and $\mathcal{T} = 100$.

## P-values computation and False Discovery Rate

PHENSIM p-value is empirically computed using the results from all simulations. Let $\mathcal{A}_{\mathcal{I}}(i)$ be the *Activity Score* computed for node $1 \leq i \leq m$ in the input simulation, and $\mathcal{A}_{\mathcal{I}_{\mathcal{R}}(r)}(i)$ be the random *Activity Score* computed for an input randomization $1 \leq r \leq \mathcal{R}$. We can say that a node alteration is not biologically relevant for the input if its probability is more significant than what might happen by chance. Therefore, if $\mathcal{A}_{\mathcal{I}_{\mathcal{R}}(r)}(i) > \mathcal{A}_{\mathcal{I}}(i)$ for most cases, we can say that the alteration is not specific for the simulated phenomena. We can synthesize this by using an empirically computed p-value as:

$$pv_{\mathcal{I}}(i) = \frac{|\{r | |\mathcal{A}_{\mathcal{I}_{\mathcal{R}}(r)}(i)| > |\mathcal{A}_{\mathcal{I}}(i)|\}|}{\mathcal{R}}. \tag{6}$$

All p-values are then corrected for multiple hypotheses using the q-value approach and given as output together with the *Activity Score* and *Average Perturbation*.

## Dealing with dependent nodes

Eq 1 implies that all input nodes are altered independently from one another. However, we might want to simulate the case where two or more nodes are dependent. Since we do not always know how this dependency might alter the LogFC distribution, we can employ a

simplified solution to address this. Indeed, we can modify the meta-pathway avoiding any changes to Eq 1.

Let $\mathcal{I}$ be the input and $\left\{ V_{i_1}, \ldots, V_{i_t} \right\} \subseteq \mathcal{I}$ the dependent nodes, where $1 \leq i_k \leq n$ and $1 \leq k \leq t \leq n$. We can create a novel node $V^*$ in the KEGG meta-pathway. Then, each edge connecting $V^* \rightarrow V_{i_k}$ is built, and its weight is assigned as $w\left( V^*, V_{i_k} \right) = v_{i_k}$, where $v_{i_k}$ is the direction of the deregulation we wish to simulate. Therefore, we can build a new input set $\mathcal{I}^*$, where all nodes are independent, as:

$$\mathcal{I}^* = \left\{ V^* = 1 \right\} \cup \mathcal{I} \setminus \left\{ V_{i_1}, \ldots, V_{i_t} \right\}.$$

This new set can be used to approximate synthetic LogFC, taking dependencies into account, without estimating how such dependencies alter Log-Fold-Changes distribution. A detailed graphical representation of the process is depicted in S6 Fig.

## Experimental procedure and benchmarking

To assess PHENSIM prediction reliability, we built a benchmark based on data published in the GEO [35] database. More in detail, we want to determine how much PHENSIM can correctly predict the biological outcomes of the up-/down-regulation of a gene in a cell line through comparisons with expression data collected before and after the alteration. Therefore, we gathered 22 GEO series of cell lines with a perturbed gene. Since these series could contain multiple perturbation experiments of different genes or in several cell lines, we obtained a total of 50 case/control sample sets. Their details are shown in S2 Table together with the name and code of the GEO series, the technology used to determine gene expression, the perturbed gene, the type of experiment (knockout, knockdown, transfection, CRISPR, etc.), whether the gene is present in KEGG pathways, and the GEO accessions of the case and control samples. Each sample set was then divided into two categories, which were analyzed differently: (i) samples whose altered gene is present in the meta-pathway (called DS1), and (ii) samples whose perturbed gene is not in the meta-pathway (called DS2). For DS1, we directly simulated the alteration of the gene using PHENSIM. For DS2, we simulated the alteration of the differentially expressed genes (DEGs) computed between cases and controls. The rationale behind this choice is that DEGs somehow represent the effect of the source alteration.

For each dataset, non-expressed genes were identified according to the experiment type: Microarray or Sequencing. For sequencing, we chose all genes with an average count of less than 10. For microarrays, we selected all genes exhibiting an average expression less than the $10^{th}$ percentile.

DEGs were computed using Limma [36] with a p-value threshold of 0.05 and a LogFC threshold of 0.6.

Each sample set was simulated as described above. Then, we compared PHENSIM predictions (up/down-regulation) with LogFC computed on the expression data. All genes showing an absolute LogFC lower than 0.6 were considered as non-altered. Finally, we assessed the results in terms of accuracy (the number of correctly predicted genes divided by the total number of genes). Furthermore, since accuracy can be influenced by class imbalance, we chose to compute Positive Predictive Value (PPV), Sensitivity, Specificity, and False Negative Rate (FNR) according to the type of alteration found in the expression data. More in detail, for altered genes (LogFC > 0.6), we want to identify upregulation and downregulation events correctly. Therefore, the True Positives (TPs) are genes predicted as upregulated with positive LogFC in the expression data. In contrast, genes predicted as downregulated with a negative

LogFC are the True Negatives (TNs). Furthermore, genes predicted as upregulated with a negative LogFC are False Positives, and downregulated genes with a positive LogFC are False Negatives. Now, we can determine the ability of PHENSIM to correctly identify upregulated genes by computing PPV and Sensitivity, while the performance regarding downregulated ones can be assessed through Specificity:

$$PPV = \frac{TP}{TP + FP},$$

$$Sensitivity = \frac{TP}{TP + FN},$$

$$Specificity = \frac{TN}{TN + FP}.$$

Concerning non-altered genes, we are interested in determining whether PHENSIM is capable of correctly identifying them. In this case, a gene that is predicted as non-altered with a LogFC < 0.6 is considered as a True Positive, while a gene indicated as altered with a LogFC < 0.6 is a False Negative. Therefore, we can estimate the rate of correctly identified non-altered genes in terms of PPV, while the FNR shows us the percentage of non-altered genes that are wrongly identified as perturbed by PHENSIM:

$$FNR = \frac{FN}{FN + TP}.$$

To compare performances with BioNSi, we ran the same simulations and computed the same metrics on the results. BioNSi requires an expression (in the range 0–9) for each gene and tracks how it changes until a steady state is reached. Therefore, a gene is up-/down-regulated if the simulated expression increases/decreases between the initial and the final state, respectively. If no change is observed, the gene is not perturbed. To run the simulation, we loaded the meta-pathway and set all genes' expression levels to 5. Next, we gave expression 9 for upregulated genes and 1 for down-regulated ones.

Moreover, since PHENSIM can extend KEGG pathways with REACTOME ones, we decided to run all tests on this extended network, comparing the results before and after the extension. However, we could not perform any comparison with BioNSi since it could not load the extended network due to its size. Finally, to quantitatively evaluate network perturbation prediction, we chose an additional dataset containing experimental measurements of protein expression changes following drug treatment in a cell line [18]. The dataset comprises 124 protein levels in a time series from 10 minutes to 67 hours (8 timepoints). The authors followed the perturbation caused by the administration of 54 drug combinations, including several gene inhibitors (MEKi, AKTi, STAT3i, SRCi, mTORi, BETi, PKCi, RAFi, and JNKi). To perform the comparison, we first gathered all drug targets from Nyman et al. [18]. Then, we simulated the alteration of their targets for each drug combination and collected the results concerning the 124 proteins. Finally, we computed the Pearson Correlation Coefficient between our predictions and the actual measurement to indicate results consistency.

All raw data, input files, and other source codes are available for download at https://github.com/alaimos/phensim/tree/master/Benchmark.

## Supporting information

**S1 Table. Summary of the comparisons between PHENSIM and BioNSi.** We computed for both software accuracy, Positive Predictive Value (PPV), Sensitivity and Specificity for genes showing altered expression, and PPV and False Negative Rate (FNR) for the non-altered ones. The sample sets were categorized based on the KEGG meta-pathway genes: DS1 contains all sample sets where the up- or down-regulated gene was in KEGG; DS2 all the remaining ones.
(XLSX)

**S2 Table. List of sample sets used for the benchmark.** Here we report a list of all sample sets used to evaluate performances of both PHENSIM and BioNSi. For each sample set, we report the GEO series from which the samples were taken together with the title and the technology used to assess expression. Furthermore, we report the altered gene, its type of alteration (Overexpression or Underexpression), and the GEO sample identifiers for both cases and controls. We also report if the gene was present and not isolated in the KEGG meta-pathway.
(XLSX)

**S3 Table. Summary of the predictions for the MAPK and NF-$\kappa$B signaling pathways.** Here, we report the most important predictions made by PHENSIM for the MAPK and NF-$\kappa$B signaling pathways. We report a set of relevant nodes for each pathway together with their perturbation, activity score, and p-value.
(XLSX)

**S1 Text. Supplementary results.**
(DOCX)

**S2 Text. Stability of the perturbation analysis.**
(DOCX)

**S1 Fig. Comparison between PHENSIM with and without REACTOME for datasets where the altered gene belongs to the meta-pathway.** Each graph reports one metric: Positive Predictive Value (PPV), Sensitivity and Specificity for genes showing altered expression, and PPV and False Negative Rate (FNR) for the non-altered ones. On the x-axis, we report PHENSIM performance with REACTOME, while on the y-axis, we present PHENSIM without REACTOME. Each dot is a dataset. The line marks the points where the two variants have the same performance.
(TIF)

**S2 Fig. Comparison between PHENSIM with and without REACTOME for datasets where the altered gene was not in the meta-pathway.** Each graph reports one metric: Positive Predictive Value (PPV), Sensitivity and Specificity for genes showing altered expression, and PPV and False Negative Rate (FNR) for the non-altered ones. On the x-axis, we report the PHENSIM performance with REACTOME, while on the y-axis, we have PHENSIM without REACTOME. Each dot is a dataset. The black line marks the points where the two algorithms have the same performance.
(TIF)

**S3 Fig. Anti-cancer effects of metformin predicted by PHENSIM.** The simulation was launched by assuming the downregulation of INS and IGF-1 and upregulation of LKB1. In S3A Fig, we show predictions related to the mTOR signaling. In S3B Fig, we show predictions related to a subset of nodes belonging to the MAPK signaling and involved in the TNF

signaling pathway. Downregulated nodes are colored in blue. Upregulated nodes are colored in red.
(TIF)

**S4 Fig. Effects of TPL2 KD and TNFα simulated by PHENSIM.** In S4A and S4B Fig are shown results obtained for TNF signaling and Apoptosis pathway, respectively, in the context of HeLa cells (chosen as representative for sensitive cell lines). In S4C and S4D Fig are shown results obtained for TNF signaling and Apoptosis pathway, respectively, in the context of RKO cells (chosen as representative for resistant cell lines). Results for CaCo-2 cells, for which PHENSIM returned a deregulation pattern like that of sensitive cell lines, are not shown. Downregulated nodes are colored in blue. Upregulated nodes are colored in red.
(TIF)

**S5 Fig. Toy example of the computation process of Eqs 3, 4 and 5.** (A) By making use of the input (upregulation of genes a and d), the algorithm prepares the starting point of the perturbation analysis ($\Delta E_{\mathcal{I}}$) by sampling from the LogFCs distributions. Such values are then propagated through the network using Eq 3 (B-E) to determine perturbation values. As soon as the steady-state is reached, we collect each gene's result. Then the process is repeated. Next, for each pathway node, we count the number of times the perturbation is positive (up-regulation), negative (down-regulation), or zero (non-expressed), and the probabilities of each state are empirically estimated (Eq 4). Finally, the activity score is established using Eq 5.
(TIF)

**S6 Fig. Manipulation of the meta-pathway when simulating dependent nodes.** We wish to simulate the upregulation of nodes $V_1$ and $V_2$ and downregulation of $V_3$. Since we know that the expression of $V_2$ and $V_3$ are dependent, we add a novel node $V^*$ which activates $V_2$ $((V^*, V_2) = 1)$ and inhibits $V_3$ $((X^*, X_2) = -1)$. Finally, we can simulate by upregulating both nodes $V_1$ and $V^*$.
(TIF)

**S7 Fig. mTORC1 and its downstream signaling pathways.** Black solid edges represent direct interaction between first neighbor nodes. Dashed edges represent indirect interactions between nodes. Red dot-dashed edges evidence scientifically validated interactions considered for PHENSIM prediction.
(TIF)

**S8 Fig. Prediction of perturbation on pathways in mammary tissue caused by Everolimus.** S8A Fig reports the top 10 list of negatively deregulated pathways, among which figured both the RNA transport and mTOR signaling pathways. In S8B **Fig** are shown predictions related to the mTOR signaling. Downregulated nodes are colored in blue. Upregulated nodes are colored in red.
(TIF)

**S9 Fig. A reconstructed model showing cellular components involved in hematopoiesis and motility of HSPCs and their downregulation mediated by exosomal-miRNAs derived from AML cells.** Black solid edges represent direct interaction between first neighbor nodes. Dashed edges represent indirect interactions between nodes. Red dot-dashed edges evidence scientifically validated interactions considered for PHENSIM prediction.
(TIF)

**S10 Fig. Prediction of perturbations caused by AML-derived exosomal miRNAs on recipient cells.** S10A and S10B Fig show the deregulation of several nodes belonging to the

Osteoclast differentiation pathway and the Cytokine-cytokine receptor interaction pathway, respectively. In **S10C Fig**, we show downregulation of c-Myb within the PI3K-Akt signaling pathway. Downregulated nodes are colored in blue. Upregulated nodes are colored in red. (TIF)

## Author Contributions

**Conceptualization:** Salvatore Alaimo, Bud Mishra, Alfredo Pulvirenti, Alfredo Ferro.

**Data curation:** Salvatore Alaimo, Rosaria Valentina Rapicavoli, Alessandro La Ferlita.

**Formal analysis:** Salvatore Alaimo, Bud Mishra, Alfredo Pulvirenti, Alfredo Ferro.

**Funding acquisition:** Alfredo Pulvirenti, Alfredo Ferro.

**Investigation:** Salvatore Alaimo, Alfredo Pulvirenti, Alfredo Ferro.

**Methodology:** Salvatore Alaimo, Bud Mishra, Alfredo Pulvirenti, Alfredo Ferro.

**Project administration:** Alfredo Pulvirenti, Alfredo Ferro.

**Resources:** Salvatore Alaimo, Rosaria Valentina Rapicavoli, Gioacchino P. Marceca, Alessandro La Ferlita, Oksana B. Serebrennikova, Philip N. Tsichlis.

**Software:** Salvatore Alaimo.

**Supervision:** Salvatore Alaimo, Alfredo Pulvirenti, Alfredo Ferro.

**Validation:** Salvatore Alaimo, Rosaria Valentina Rapicavoli, Gioacchino P. Marceca, Alessandro La Ferlita, Oksana B. Serebrennikova, Philip N. Tsichlis, Bud Mishra, Alfredo Pulvirenti.

**Visualization:** Salvatore Alaimo, Rosaria Valentina Rapicavoli, Gioacchino P. Marceca, Alessandro La Ferlita.

**Writing – original draft:** Salvatore Alaimo, Rosaria Valentina Rapicavoli, Gioacchino P. Marceca, Alessandro La Ferlita, Oksana B. Serebrennikova, Philip N. Tsichlis, Bud Mishra, Alfredo Pulvirenti, Alfredo Ferro.

**Writing – review & editing:** Salvatore Alaimo, Rosaria Valentina Rapicavoli, Alfredo Pulvirenti, Alfredo Ferro.

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
