## [Decision Letter · Decision Letter 0]

20 Jan 2021

Dear Prof. Ferro,

Thank you very much for submitting your manuscript "PHENSIM: Phenotype Simulator" for consideration at PLOS Computational Biology.

As with all papers reviewed by the journal, your manuscript was reviewed by members of the editorial board and by several independent reviewers. In light of the reviews (below this email), we would like to invite the resubmission of a significantly-revised version that takes into account the reviewers' comments.

We cannot make any decision about publication until we have seen the revised manuscript and your response to the reviewers' comments. Your revised manuscript is also likely to be sent to reviewers for further evaluation.

Sincerely,

Christos A. Ouzounis

Associate Editor

PLOS Computational Biology

Jason Haugh

Deputy Editor

PLOS Computational Biology

Reviewer's Responses to Questions

**Comments to the Authors:**

Reviewer #1: This manuscript describes a new algorithm for predicting the effects of altered genes, miRNA, and metabolites on KEGGS curated pathways.

This addresses a particularly challenging problem in system biology of predicting the impact of directed (through drug treatment or experiment) or disease dysregulation of pathways. The algorithm is based on the rational of simulating log fold expression changes and its propagation through the network through a weighted sum of up-stream and down-stream connected nodes.

The comparison to BioNSi demonstrates that PHESIM has comparable and even improved performance.

My main criticism is that there should be a more quantitative evaluation of the prediction of network perturbations. While the algorithm returns activity score for each node the overall evaluation of the network perturbation relative to expected is left for expert interpretation. In the four biological examples the authors perform a qualitative assessment of the PHESIM performance based on prior biological knowledge. Admittedly, this is a difficult assessment. One suggestion is to evaluate the algorithm on additional experimental datasets with experimental measurements of gene expression changes following drug treatment or other gene perturbation. For example, PLoS Comput Biol . 2020 Jul 15;16(7):e1007909. doi: 10.1371/journal.pcbi.1007909,

or 10.1371/journal.pcbi.1003290.

The authors should also discuss the algorithm’s performance in pathways with feedback loops. Specifically, what is the stability of the scores in feedback loops? do they tend to oscillate between activation or suppression? Is the final score sensitive to the number of perturbations performed?

Few additional comments:

- On page 6, a better description of the test data sets is needed. What is “…22 series further divided into 50 case/control sets”? Are these 50 experiments of gene KO experiments? What are the types of perturbations, shRNA, CRISPR?

- Line 124, page 6, are BioNSi numbers 0.0640, 0.0735? is there extra 0 after decimal?. If true this means BioNSi has no accuracy in these test sets.

- It will be helpful to provide additional descriptions of the performance criteria in the context of this work. For example, how is PPV and False Negative Rates computed in this context?

- In the method section, specifically for PHENSIM simulation step and Activity score, it will be helpful to add a graph figure to illustrate the computation (eq. 3 and 4). Similar to Figure S7.

- In equation 3, how does the algorithm handle the condition when the sum in the denominator is equal to 0 (i.e the sum of weights of upstream and downstream nodes)?

- The web app is a nice feature of this work. However, the submission UI requires better description. What are the various input fields (nodes) mean? Node is a graph theory term not biological. What is the difference between KO nodes and under-expressed nodes? Are all fields required or only a subset?

Reviewer #2: This is an interesting piece of work, which targets a topic with high prospective importance for the bioinformatic community and the system biology research in general. Genome scale models will be more pressingly requested in the coming years, so personally I welcome any work in this field.

The tool has been implemented so as to enable easy use and it integrates seamlessly and efficiently the provenance source of KEGG Pathways. However, I think that at its current status, the results that can be generated are of limited importance, primarily because of the multi-tiered architecture of biological networks and the extensive organisational and thermodynamic coupling that governs them.

In that sense I believe that this work would be largely benefit from an effort to integrate more provenance source for cellular networks like Reactome for instance that the authors cite, or even more genome-scale modelling efforts like BioCyc, or network information that could be extracted from massive interaction network data.

Another improvement that I would consider largely necessary primarily because it is the advantage of KEGG Pathways that the tool is relying on, is the addition of more supported organisms, as in this way probably the tool would find broader applicability in research communities seeking for similar tools (plant or microbial biologists) than the current ones that the tool targets.

Also regarding the topic of super pathway modelling there have been seminal efforts in the past that have provided solutions to that (tools like KEGGConverter or KENeV for instance. It is important that the authors cite some of these effort and explain briefly their point of novelty to prior efforts.

Lastly, it is important that the authors ensured support of interoperability of the networks that their tool is building by supporting export to an XML format like SBML, KGML or RDF format for instance. This could provide larger visibility for the tool and extend its widespread use.

**Have all data underlying the figures and results presented in the manuscript been provided?**

Reviewer #1: Yes

Reviewer #2: Yes

PLOS authors have the option to publish the peer review history of their article (what does this mean?). If published, this will include your full peer review and any attached files.

Reviewer #1: No

Reviewer #2: No
---

## [Decision Letter · Decision Letter 1]

12 May 2021

Dear Prof. Ferro,

We are pleased to inform you that your manuscript 'PHENSIM: Phenotype Simulator' has been provisionally accepted for publication in PLOS Computational Biology.

Best regards,

Christos A. Ouzounis

Associate Editor

PLOS Computational Biology

Jason Haugh

Deputy Editor

PLOS Computational Biology

Reviewer's Responses to Questions

**Comments to the Authors:**

Reviewer #1: None

Reviewer #2: In the revised form, the authors have efficiently addressed all points raised from my side. Taken together, all revisions made, have improved significantly the merit of this work, so from my side, I consider the manuscript suitable for publication.

**Have the authors made all data and (if applicable) computational code underlying the findings in their manuscript fully available?**

Reviewer #1: Yes

Reviewer #2: Yes

PLOS authors have the option to publish the peer review history of their article (what does this mean?). If published, this will include your full peer review and any attached files.

Reviewer #1: No

Reviewer #2: **Yes: **Aristotelis Chatziioannou

---

## [Editor Report · Acceptance letter]

3 Jun 2021

PCOMPBIOL-D-20-01854R1 

PHENSIM: Phenotype Simulator

Dear Dr Ferro,

I am pleased to inform you that your manuscript has been formally accepted for publication in PLOS Computational Biology. Your manuscript is now with our production department and you will be notified of the publication date in due course.

With kind regards,

Katalin Szabo
